# *Staphylococcus aureus* Protection-Related Type 3 Cell-Mediated Immune Response Elicited by Recombinant Proteins and GM-CSF DNA Vaccine

**DOI:** 10.3390/vaccines9080899

**Published:** 2021-08-13

**Authors:** Kamila R. Santos, Fernando N. Souza, Eduardo M. Ramos-Sanchez, Camila F. Batista, Luiza C. Reis, Wesley F. Fotoran, Marcos B. Heinemann, Hiro Goto, Magnus Gidlund, Adriano F. Cunha, Angélica Rosa Faria, Hélida M. Andrade, Andrey P. Lage, Mônica M. O. P. Cerqueira, Alice M. M. P. Della Libera

**Affiliations:** 1Veterinary Clinical Immunology Research Group, Departamento de Clínica Médica, Faculdade de Medicina Veterinária e Zootecnia, Universidade de São Paulo, São Paulo 05508-270, Brazil or fernando.nogueira.souza@academico.ufpb.br (F.N.S.); camilafre@gmail.com (C.F.B.); dellalibera@usp.br (A.M.M.P.D.L.); 2Programa de Pós-Graduação em Ciência Animal, Universidade Federal da Paraíba, Areia 58397-000, Brazil; 3Laboratório de Soroepidemiologia e Imunobiologia, Instituto de Medicina Tropical, Faculdade de Medicina, Universidade de São Paulo, São Paulo 05403-000, Brazil; eduardors22@hotmail.com (E.M.R.-S.); luizacreis@gmail.com (L.C.R.); hgoto@usp.br (H.G.); 4Departamento de Salud Publica, Facultad de Ciencias de La Salud, Universidad Nacional Toribio Rodriguez de Mendoza de Amazonas, Chachapoyas 01000, Peru; 5Laboratório de Genética, Divisão de Biologia, Instituto Butantan, São Paulo 05503-900, Brazil; wesleylfw@hotmail.com; 6Departamento de Medicina Veterinária Preventiva e Saúde Animal, Faculdade de Medicina Veterinária e Zootecnia, Universidade de São Paulo, São Paulo 05508-270, Brazil; marcosbryan@usp.br; 7Departamento de Medicina Preventiva, Faculdade de Medicina, Universidade de São Paulo, São Paulo 01246-903, Brazil; 8Departamento de Imunologia, Instituto de Ciências Biomédicas, Universidade de São Paulo, São Paulo 05508-900, Brazil; gidlundm@usp.br; 9Departamento de Tecnologia e Inspeção de Produtos de Origem Animal, Escola de Veterinária, Universidade Federal de Minas Gerais, Belo Horizonte 31270-010, Brazil; adrianofcunha@hotmail.com.br (A.F.C.); monicapinhocerqueira@gmail.com (M.M.O.P.C.); 10Departamento de Parasitologia, Instituto de Ciências Biológicas, Universidade Federal de Minas Gerais, Belo Horizonte 31270-901, Brazil; angelicarosafaria@gmail.com (A.R.F.); helidandrade@gmail.com (H.M.A.); 11Laboratório de Parasitologia Clínica, Faculdade de Ciências Farmacêuticas, Universidade Federal de Alfenas, Alfenas-MG 37130-000, Brazil; 12Departamento de Medicina Veterinária Preventiva, Escola de Veterinária, Universidade Federal de Minas Gerais, Belo Horizonte 31270-010, Brazil; aplage@vet.ufmg.br or

**Keywords:** vaccine, *Staphylococcus aureus*, T cell response, mastitis, bovine

## Abstract

*Staphylococcus aureus* mastitis remains a major challenge for dairy farming. Here, 24 mice were immunized and divided into four groups: G1: control; G2: Granulocyte Macrophage Colony-Stimulating Factor (GM-CSF) DNA vaccine; G3: F0F1 ATP synthase subunit α (SAS), succinyl-diaminopimelate (SDD), and cysteinyl-tRNA synthetase (CTS) recombinant proteins; and G4: SAS+SDD+CTS plus GM-CSF DNA vaccine. The lymphocyte subpopulations, and the intracellular interleukin-17A (IL-17A) and interferon-γ production in the draining lymph node cells were immunophenotyped by flow cytometry. The immunophenotyping and lymphocyte proliferation was determined in spleen cells cultured with and without *S. aureus* stimulus. Immunization with *S. aureus* recombinant proteins generated memory cells in draining lymph nodes. Immunization with the three recombinant proteins plus GM-CSF DNA led to an increase in the percentage of IL-17A^+^ cells among overall CD44^+^ (memory), T CD4^+^, CD4^+^ T CD44^+^ CD27^−^, γδ TCR, γδ TCR^+^ CD44^+^ CD27^+^, and TCRVγ4^+^ cells. Vaccination with *S. aureus* recombinant proteins associated with GM-CSF DNA vaccine downregulated T_H_2 immunity. Immunization with the three recombinant proteins plus the GM-CSF DNA led to a proliferation of overall memory T, CD4^+^, and CD4^+^ TEM cells upon *S. aureus* stimulus. This approach fostered type 3 immunity, suggesting the development of a protective immune response against *S. aureus*.

## 1. Introduction

Bovine mastitis is the disease with the greatest global impact on dairy farming, causing a pronounced reduction in milk production and quality. Several etiologic agents are implied in mastitis, but *Staphylococcus aureus* is considered one of the most prevalent mastitis pathogens. It is an extremely critical pathogen due to its pathogenicity, contagiousness, and refractoriness to antimicrobial treatment [1,2], as well as critical issues related to food security and public health [3]. Therefore, to decrease the impact of *S. aureus* in dairy farming, alternative strategies to control mastitis should be evoked, such as the development of an effective vaccine.

*Staphylococcus aureus* pathogenicity is a complex issue; thus, a single antigen may not confer protective immunity [4]. Most studies have focused on *S. aureus* virulence factors for developing vaccines against *S. aureus* intramammary infections (IMIs) in dairy cows, but this initiative has been unsuccessful so far. Here we used an innovative approach focusing on the animals that apparently had overcome the infection by developing successful anti-pathogen host immune responses. We intended to take advantage of the knowledge derived from detailed analysis and characterization of the immune response that favored the host to identify promising epitopes for immunization [1,2,5,6]. Using this approach, we previously searched *S. aureus*-derived proteins likely related to protection [2]. Based on these antigenic proteins revealed by a serum immunoproteomics approach, we obtained three candidate proteins: F0F1 ATP synthase subunit α (SAS), succinyl-diaminopimelate (SDD), and cysteinyl-tRNA synthetase (CTS) [2].

Some vaccine candidate antigens induced a robust humoral immunity, but they have failed so far [6,7]. Thus, there is growing evidence that *S. aureus* vaccine strategies aimed at eliciting both appropriate long-lived B and T cell responses should be spotlighted [5,8,9]. Therefore, a more comprehensive characterization of T cell immunity [9] is proposed to characterize the immune response induced by the proteins mentioned above and to analyze it in the context of protective immunity.

T cell immunity has been increasingly reckoned as a central player in host-pathogen interaction, and T cell-derived interleukin (IL)-17 as a cornerstone of protection in *S. aureus* infections [8,10,11,12,13]. In this regard, the use of a DNA vaccine containing Granulocyte Macrophage Colony-Stimulating Factor (GM-CSF) appears as a powerful immunoadjuvant as it increases the lymphoproliferative response and IL-17 cytokine production [14]. Furthermore, the administration of a recombinant human GM-CSF alone was seen to reduce the rate of new IMIs by 46.7% following a challenge with *S. aureus* [15].

Aiming for an effective vaccine for *S. aureus* bovine IMIs, in the present study we propose evaluating the immune response profile triggered by three recombinant *S. aureus* vaccine candidate antigens (SAS, SDD, and CTS) in association with GM-CSF DNA vaccine in a mouse model.

## 2. Materials and Methods

### 2.1. Ethical Statement

This study was approved by the Animal Research Ethics Committee of the Universidade de São Paulo, Brazil, under protocol number 6276100519.

### 2.2. Production of S. aureus Recombinant Proteins

The sequence of amino acids corresponding to the SAS (Genbank gi|446897629), SDD (Genbank gi|486621908), and CTS (Genbank gi|257275504) *S. aureus* antigens were codon-optimized for *Escherichia coli* expression. Genes were commercially synthesized by Genscript, Piscataway, NJ, USA. The produced synthetic genes were cloned into the pUC57 vector and then sub-cloned into the pET28a expression vector [16]. Recombinant plasmids were used to transform the expression strain *E. coli* BL21-Star™(DE3) as previously described [17] with modifications. Briefly, transformed *E. coli* BL21-Star™(DE3) were selected on kanamycin plates. An overnight bacterial culture of three colonies containing the respective expression plasmid was grown in Luria-Bertani medium (LB) in the presence of kanamycin (0.05 mg mL**^−^**^1^) until an optical density (OD) of 0.4 at 600 nm, on a rotary shaker at 37 °C. Then, isopropyl-ß-d-1-thiogalactopyranoside (IPTG, Sigma) was added to the culture to a final concentration of 0.4 mM, and the induced cultures were grown for 4 h. Cells were ruptured by ultrasound sonication on ice and debris removed by centrifugation (20,000× *g*, at 4 °C, for 30 min). The recombinant proteins were purified using Ni Sepharose High Performance immobilized metal ion affinity chromatography columns (HisTrap Hp, cat. n. GE17-5248-02, GE Healthcare, Logan, UT, USA), attached to an ÄKTA Pure (GE Healthcare, Chicago, IL, USA), under denaturing conditions according to the manufacturer’s instructions. The sizes and purity of the *S. aureus* recombinant proteins were assessed by SDS-PAGE, as shown in Appendix A.

### 2.3. Production of GM-CSF DNA Vaccine

The amino acid sequence of the GM-CSF (Genbank, gi|145301581) was forwarded to FastBio (Ribeirão Preto, Brazil) for optimization for eukaryotic cells and to synthesize the respective gene. The produced synthetic genes were cloned in pUC57 vector and then were sub-cloned into the pCI-neo mammalian expression vector insert (Promega Incorporation, Madison, WI, USA). *E. coli* DH5α was transformed with pCI-GM-CSF, and immunization plasmids were purified using the ZR plasmid Gigaprep Kit (Zymo Research, Irvine, CA, USA), according to the manufacturer’s instructions. The expression of the GM-CSF gene was checked and confirmed in the RAW 264.7 cells transfected with our GM-CSF DNA vaccine by qPCR (data not shown).

### 2.4. Liposome Preparation and Entrapment of Plasmid DNA

We used a cationic liposome to efficiently deliver the GM-CSF DNA plasmid DNA, which has been associated with induced strong humoral and cell-mediated responses [18]. Briefly, liposomes were prepared using dimethyl-di-octadecyl-ammonium (DDAB), cholesterol (molar ratio 1:4) and 1,2-distearoyl-sn-glycero-3-phosphoethanolamine-*N*-[amino(polyethylene glycol)-2000 (DSPE + PEG2000), 5% of the total lipids used. First, 1 mL of chloroform was used to dissolve this lipid solution and then left under a constant N2 flow to evaporate chloroform and develop a phospholipid film on the tube walls. This film was then retained under vacuum for at least 1 h to eliminate any remaining chloroform. The film was rehydrated in 5 mM Tris-HCl (pH 7.5) at 60 °C for 1 h and vigorously stirred every 10 min. The obtained opaque solution was then exposed to sonication at high energy until the solution became almost fully transparent. Then, the solution was centrifuged for 1 h at 100,000× *g*. Any residual pellet was removed, and the supernatant containing unilamellar lipid vesicles was used to form the liposome. The genetic material was then added to the liposomes in a molar stoichiometry of 8 nM DDAB for each 1 µg of nucleic acids, the molarity being approximately 0.2 pmol.

### 2.5. Animals and Immunization

Twenty-four C57BL/6J mice (six-week-old, female) were purchased from the Centro de Bioterismo of Faculdade de Medicina, Universidade de São Paulo. After three weeks of adaptation, the animals (nine weeks old) were divided into four groups (six animals per group) and immunized as defined in Figure 1. Here, 50 µg of pCI-GM-CSF plasmid DNA resuspended in 100 µL of sterile liposomal formulation were administered subcutaneously under the skin of the interscapular area near the draining lymph nodes. The recombinant SAS, SDD, and CTS proteins (20 µg each protein) in 100 µL of saponin adjuvant (Quil-A**^®^**, cat. n. 8047-15-2, Invitrogen, San Diego, CA, USA) were administered intramuscularly in the deltoid muscles (Figure 1).

On day 44 [14 days after the last administration of the recombinant protein(s)], the animals were anesthetized intramuscularly with xylazine (10 mg kg**^−^**^1^) and ketamine (50 mg kg**^−^**^1^) solution and euthanized by cervical dislocation. Then, the draining lymph nodes (accessory axillary and proper axillary) and spleen were removed to obtain the cells for further immunophenotyping and lymphocyte proliferation evaluation, respectively. The accessory axillary and proper axillary lymph nodes from the side of vaccination were pooled in each animal.

### 2.6. Obtaining the Spleen and Lymph Nodes Cells

The spleens or lymph nodes of donor mice were aseptically removed, mechanically disrupted, and homogenized by syringe plunger into a cell strainer (cat. n. Z742102, Sigma Aldrich, St. Louis, MO, USA) on the top of a 50 mL tube with 10 mL of RPMI 1640 medium (cat. n. R7638, Sigma Aldrich, St. Louis, MO, USA) supplemented with 5% heat-inactivated fetal bovine serum (Gibco, Waltham, MA, USA), 100 U mL-1 penicillin, 100 μg mL**^−^**^1^ streptomycin, and Fungizone 0.25 μg mL**^−^**^1^ (cat. n. 15240-096, Gibco, Waltham, MA, USA) (complete medium).

The spleen and lymph node cells were centrifuged at 380× *g*, 10 min, at 4 °C. Thereafter, the hypotonic lysis of erythrocytes was performed by adding 1000 μL of 0.2% NaCl for 20 s; then, isotonicity was restored by adding 1000 μL of 1.6% NaCl. Cells were further centrifugated and suspended in a complete medium. Cell viability was checked with the trypan blue exclusion test (cat. n. T8154-100ML, Sigma Aldrich, St. Louis, MO, USA).

### 2.7. Immunophenotyping of Ex-Vivo Lymph Nodes Cells

The lymph node cells were phenotyped using 0.5 µL of each of the fluorescent-conjugated monoclonal antibodies (mAbs; Table 1) for cell surface CD4 (clone GK1.5), CD8 (clone 53-6.7), CD19 (clone 1D3), γδ TCR (clone GL3), TCRVγ4 (clone Gl2), CD44 (clone IM7), and CD27 (clone LG.3A10) by incubating the cells for 30 min at room temperature in the dark. After the incubation period, the cells were washed with 0.01 M phosphate-buffered saline, pH 7.4 (PBS), and spun at 250× *g* at 4 °C for 8 min. Further, the intracellular production of IL-17A and interferon (IFN-γ) were determined after by fixing and permeabilizing the cells as previously described [19], staining with 0.5 µL of fluorescein-conjugated anti-mouse IL-17 and anti-mouse IFN-γ antibodies (Table 1), respectively, and incubating for one hour at 4 °C. After the incubation period, 500 μL of the permeabilization solution (PBS + 0.1% saponin + 0.09 azide + 1% heat-inactivated fetal bovine serum) was added and the samples were centrifuged at 250× *g* at 4 °C for 8 min. Then, the samples were resuspended in 300 μL PBS with 1% SFB and analyzed by flow cytometry (BD LSRFortessa^TM^ X-20 flow cytometer, Becton Dickinson Immunocytometry System^TM^, San Diego, CA, USA). Here, 100,000 events were acquired per sample. Flow Jo Tree Star software (FlowJo—Treestar 10.5.3 for Windows, Tree Star Inc., Ashland, OR, USA) was used to analyze the data. Non-stained control and single-stained samples as compensation controls, negative control stained with conjugated isotype control antibody, and cells stained with fluorescence minus-one (FMO) controls were included. Doublets were excluded using forward scatter (FSC) area versus FSC height.

### 2.8. Preparation of S. aureus Inoculum

An udder adapted *S. aureus* isolate originated from a case of chronic subclinical IMI was used [17]. *Staphylococcus aureus* (spa typing t605) inoculum was prepared as previously described [20] with minor modifications. Bacteria were resuspended in RPMI-1640 (cat. n. R7638, Sigma Aldrich, St. Louis, MO, USA) with 10% heat-inactivated fetal bovine serum (Cultilab, Campinas, Brazil). To determine the bacteria concentration in the inoculum, an aliquot of the bacterial suspension was further cultured on BHI agar plates in dilution series, and colony numbers (colony forming units mL**^−^**^1^) were determined. Then, the bacterial concentration was adjusted to the final inoculum dose (2 × 10^8^ staphylococci mL**^−^**^1^) to obtain a multiplicity of infection (MOI) = 10. The inoculum was heat-inactivated at 60 °C for one hour, after which 100 µL of the solution was plated on a plate containing blood agar and incubated at 37 °C for 24 h to confirm inactivation.

### 2.9. Immunophenotyping and Lymphocyte Proliferation of Cultured Spleen Cells

For lymphocyte activation of spleen cells, 2 × 10^5^ viable cells per well were cultured at 37 °C and 5% CO_2_ for 96 h in 96-wells flat-bottom plates with or without (control) 10 µL of heat-inactivated *S. aureus* (2 × 10^8^ CFU mL**^−^**^1^). After the incubation period, the cells were harvest from the 96-well plates and transferred to 5 mL tubes, round bottom, 12 × 75 mm, for flow cytometry, and centrifuged at 250× *g* at 4 °C for 8 min. Then, the supernatant was collected and stored for cytokines measurement, and the cells were phenotyped using fluorescent-conjugated mAbs (Table 1).

Furthermore, the intracellular production of interleukin (IL)-17A and interferon (IFN)-γ, in addition to the antigen-specific in vitro lymphocyte proliferation marker [19], were determined. Ki67 is a nuclear protein that acts in the regulation of the cell division process. This protein is expressed during all active phases of cell division; thus, it has been widely used to evaluate cell proliferation [19]. After centrifugation, the cells were fixed, and further permeabilized [19]. Afterward, 0.5 µL of fluorescent-conjugated anti-mouse IL-17A and IFN-γ (Table 1), and 10 μL of diluted (2 μL of ki67 diluted in 198 μL of permeabilization solution) rabbit anti-Ki67 antibody solution (cat n. ab15580, abcam, Cambridge, UK), was added and incubated for 1 h at 4 °C. After the incubation period, 500 μL of the permeabilization solution was added, and the samples were centrifuged at 250× *g* at 4 °C for 8 min. The supernatant was discarded, and 1 μL of fluorescein isothiocyanate (FITC)-conjugated goat anti-rabbit IgG H&L secondary antibody (cat n. ab6717, abcam, Cambridge, UK) was added to tubes and incubated again for 30 min at room temperature in the dark. After incubation, 500 μL of the permeabilization solution was added, and the samples were centrifuged at 250× *g* at 4 °C for 8 min. After centrifugation, the samples were resuspended in 300 μL of PBS with 1% SFB and analyzed by flow cytometry (BD LSRFortessa^TM^ X-20 flow cytometer, Becton Dickinson Immunocytometry System^TM^, San Diego, CA, USA). For this assay, 100,000 events were examined in each sample. Flow Jo Tree Star software (FlowJo—Treestar 10.5.3 for Windows, Tree Star Inc., Ashland, OR, USA) was used to analyze the data. Non-stained control and single-stained samples were prepared as compensation controls. Negative control samples were stained with conjugated isotype control antibodies. In addition, cells were stained with fluorescence minus-one (FMO) controls. Doublets were excluded using forward scatter (FSC) area versus FSC height.

### 2.10. Cytokines Measurement

The production of cytokines IL-2, IL-4, IL-6, IL-10, IFN-γ, IL-17A, and tumor necrosis factor (TNF)-α was determined in the supernatant of the cultured spleen cells with or without *S. aureus* stimulation, using the BD Cytometric Bead Array Mouse T_H_1, T_H_2, and T_H_17 Cytokine (cat. n. 560485, BD Bioscience^TM^, San Jose, CA, USA), and using a flow cytometer (BD LSRFortessa^TM^ X-20 flow cytometer, Becton Dickinson Immunocytometry System^TM^, San Diego, CA, USA), as per the manufacturer’s instructions. The FCAP Array^TM^ v3.0 software (Softflow^TM^, Pécs, Hungary) was used to analyze the data.

### 2.11. Statistical Analysis

Statistical analysis was performed using GraphPad Prism 9 (GraphPad Software, Inc., San Diego, CA, USA). Data are presented as percentages of distinct T cell populations that produced IL-17A or IFN-γ (percentage of stained cells) and geometric mean fluorescence intensity (GMFI), which provides an accurate measurement of the brightness of stained cells. The GMFI of 17A or IFN-γ was determined among IL-17^+^ or IFN-γ+, indicating the intensity of cytokine production per cell that produced the respective cytokine. To assess the percentage of each lymphocyte subpopulation, the percentage of proliferative cells (ki67^+^), as well as the percentage and the GMFI of IL-17A or IFN-γ, a stimulation index (SI) was calculated by dividing the percentage of positive cells upon *S. aureus* stimulation per percentage of positive cells under unstimulated control condition [21,22,23,24,25]. The data were first tested for normality of the distribution using the Kolmogorov–Smirnov and Shapiro–Wilk tests. If the distribution was normal, the data of the experimental groups were subjected to a one-way ANOVA analysis followed by the Tukey test. Variables with non-parametric distributions were analyzed using the Kruskal–Wallis test followed by the Dunn’s test. Results are reported as mean + standard error of the mean. *p* ≤ 0.05 was considered significant unless otherwise indicated.

## 3. Results

### 3.1. Immunization with S. aureus Recombinant Proteins Generate Memory Cells in Draining Lymph Nodes

CD44 is a prominent activation marker that differentiates memory and effector T cells from their naïve counterparts [26]. A higher percentage of CD44^+^ cells in draining lymph nodes was found in animals that just received *S. aureus* recombinant proteins (data not shown). The memory T cells were segregated into two distinct populations: a CD44^high^ CD62L^low^ that exert a rapid effector function (so-called effector memory, TEM), which have poor proliferative capacity; and a CD44^high^ CD62L^high^ population with no immediate effector function (central memory, TCM) that possess high proliferative potential [27,28]. Those animals that only received the combination of *S. aureus* recombinant proteins showed an enhanced percentage of T CD4^+^ CD44^+^ CD27^+^, i.e., TCM T CD4^+^ cells (Figure 2A), but a reduced percentage of T CD8^+^ CD44^+^ CD27**^−^**, i.e., effector T CD8^+^ cells (Figure 2B). The percentage of CD4^+^ CD44^+^ CD27^−^ (*p* = 0.27) and CD8^+^ CD44^+^ CD27^+^ (*p* = 0.75) showed no significant difference.

### 3.2. TCRVγ4^+^ Cells Are Responsible for the Greatest Contribution for Type 1 and 3 Cell-Mediated Immunities in the Draining Lymph Nodes

An emerging concept is that unconventional T cells, such as γδ T cells, may confer protective immunity against *S. aureus* [13,29]. Here, we observed in the draining lymph nodes that γδ TCR cells and their subpopulation, TCRVγ4^+^ cells, were associated with the highest percentage of IL-17A^+^ cells (*p* ≤ 0.0001; Figure 3), highlighting the critical importance of this lymphocyte population in triggering type 3 cell-mediated immunity [13,30]. Furthermore, the TCRVγ4^+^ cell population exhibited a higher percentage of IFN-γ^+^ cells than all αβ T lymphocyte subpopulations, while γδ TCR^+^ cells just differed from T CD8^+^ lymphocytes (Figure 3).

### 3.3. Immunization with S. aureus Recombinant Proteins Associated with GM-CSF DNA Vaccine Drives Type 3 Recall Immunity That Largely Relies on γδ T and TCRVγ4^+^ Cells

There is increasing evidence that type 3 immunity mediated by cells that produce IL17-A and IL-17F is crucial for mammary gland protective immunity [7], and consequently a promising target for vaccine development. Observing that the three *S. aureus* recombinant antigens plus the GM-CSF DNA vaccine induced an appropriate immune response at the draining lymph node, we proceeded to evaluate the systemic recall immune response using splenic cells from immunized mice stimulated in vitro with *S. aureus*. No effect of immunization on the induction of B cell population (CD19^+^) (*p* = 0.21) response was revealed (data not shown). To quantify the intensity of response upon *S. aureus* stimulation, T cell responses are normalized by dividing the percentage of positive cells upon *S. aureus* stimulation by the percentage of positive cells under unstimulated control conditions, thereby creating a stimulation index. Thus, when in vitro cultures of spleen cells were stimulated with *S. aureus* from vaccinated mice with the three recombinant proteins plus the GM-CSF DNA vaccine, an increase in the percentage of IL-17A^+^ cells among overall CD44^+^ (memory) cells (35.34%), T CD4^+^ cells (54.36%), CD4^+^ TEM (CD44^+^ CD27^−^) cells (142.87%), and TCRVγ4^+^ cells (56.75%) were found (Figure 4). This effect was due almost to no significant impact on the IL-17A^+^ among the distinct T cells subsets in spleen cell culture under *S. aureus* stimulus compared with unstimulated conditions in animals from unvaccinated, GM-CSF DNA vaccine, or those vaccinated with the *S. aureus* recombinant proteins alone (*p* ≤ 0.05); while mice immunized with the three-antigens *S. aureus* recombinant proteins associated with the GM-CSF DNA vaccine generated a pronounced enhancement in the percentage of IL-17A^+^ cells in all abovementioned T cell populations, primarily on γδ T cells and TCRVγ4^+^ cells. Nevertheless, no vaccine effect on GMFI of IL-17A production or type 1 (IFN-γ^+^) immune cells was observed.

### 3.4. TCRVγ4^+^ Cells and T CD8^+^ Cells Are the Major Contributors for Type 1 Immunity, While γδ TCR Cells Mainly Support Type 3 Immunity in Splenocyte Cell Culture

Here, we observed that both TCRVγ4^+^ cells and T CD8^+^ cells are the primary source of both IFN-γ, followed by γδ TCR cells (Figure 5). On the other hand, γδ TCR cells represent the major producer of IL-17A, followed by TCRVγ4^+^ cells and T CD8^+^ cells (Figure 5). The T CD4^+^ cells marginally produce IFN-γ and IL-17A, although the major population found splenocyte cell culture in both unstimulated and *S. aureus* stimulated conditions (Figure 5).

### 3.5. Immunization of Mice with S. aureus Recombinant Proteins Associated with GM-CSF DNA Vaccine DownregulatesTh2 Immunity

The cytokine measurements in the supernatant of the unstimulated (control) and *S. aureus* stimulated cultured spleen cells are shown in Appendix A. In the present study, the IL-4 concentration in the supernatant of the splenocyte culture upon *S. aureus* stimulation was lower in animals vaccinated with the GM-CSF DNA vaccine (*p* = 0.08), those immunized with the three-recombinant antigens (*p* = 0.03), and immunized with the three-recombinant antigens together with the GM-CSF vaccine (*p* = 0.06) than compared with unvaccinated control animals (*p* = 0.50). Moreover, the IL-4 concentration in the supernatant of the splenocyte culture upon *S. aureus* was higher in the non-immunized control mice than in those animals that received the three *S. aureus* recombinant proteins alone (*p* = 0.01).

No difference in the concentration of TNF-α and IL-6 in the supernatant of the cell culture among groups, or between unstimulated and *S. aureus* stimulated conditions, were detected. The concentrations of IL-10, IL-17A, and IFN-γ were below the limit of the detection. In addition, *S. aureus* stimulation led to a decrease in the concentration of IL-2 in the supernatant of splenocyte culture only in mice that received the GM-CSF DNA vaccine together with the recombinant proteins (*p* = 0.05).

### 3.6. The Combination of S. aureus Recombinant Proteins and GM-CSF Vaccine Induces a Higher Proliferation of Memory T Cells and T CD4^+^ Cells

In the present study, when we compared the percentage of proliferative cells between *S. aureus* stimulated and unstimulated conditions within groups, we observed that animals that received the *S. aureus* recombinant proteins plus GM-CSF DNA vaccine showed an increase in the percentage of proliferative (ki67^+^) overall memory T cells (CD44^+^; *p* = 0.02), CD4^+^ cells (*p* = 0.02) and CD4^+^ TEM (CD44^+^ CD27**^−^**; *p* = 0.01) when stimulated with *S. aureus*, which was not sustained in other experimental groups as shown in Appendix A.

## 4. Discussion

Vaccination strategies may provide memory T cells to fight developing pathogens in the host and respond quickly upon reencounter with the pathogen [28,30,31,32]. Within this context, a hallmark finding of our study is the high production of IL-17A by diverse T cell memory cell subsets using the combination of three *S. aureus* recombinant proteins and GM-CSF DNA vaccination when stimulated with *S. aureus*. Further, we must highlight the increased production of IL-17A by the TCRVγ4^+^ cell population when stimulated with *S. aureus*. In the present study, we demonstrated for the first time that the combination of *S. aureus* CTS, SDD, and SAS recombinant proteins associated with a GM-CSF DNA plasmid DNA vaccine foster type 3 cell-mediated immunity that turns these outcomes into a promising added value.

We should emphasize the strategy to identify three recombinant proteins used in the present study using the serum immunoproteomics approach in the previous study [2]. In this latter study, we searched the antibodies produced by dairy cows which were exposed to *S. aureus* but remained healthy, hypothesizing that the antibodies repertoire directed against these specific *S. aureus* proteins contributed to mastitis resistance. The first *S. aureus* protein, the so-called SAS protein, plays a pivotal role in dictating biofilm growth and structure [33], enhances tolerance of *S. aureus* towards some host antimicrobial peptides of the innate immunity [34], inhibits pro-inflammatory cytokine production by monocyte-derived macrophages and myeloid-derived suppressor cells, and renders *S. aureus* more resistant to the bactericidal activity of monocyte-derived macrophages [33]. The second *S. aureus* SDD protein is critical for bacteria survival and proliferation. It is involved in protein synthesis and constructing the peptidoglycans of the bacterial wall [35]. Lastly, *S. aureus* CTS is an aminoacyl-tRNA enzyme that catalyzes the binding of a specific amino acid to tRNA, attaching the amino acid to other molecules, and is therefore involved in protein synthesis [35]. Altogether, although molecular functions for these proteins are not exactly known, we consider it reasonable to speculate that stimulating the host immune response against these proteins is the precise strategy to protect animals from *S. aureus* infections.

In this scenario, although most available vaccines’ success relies on antibody-mediated immunity, it is conceivable that humoral immunity alone may not be sufficient to protect against *S. aureus* infections [5,6] fully. Incidentally, protective immunity against recurrent *S. aureus* skin and soft tissue infection in a human was associated with both antibody and T-cell mediated (i.e., IL-17) immunities, suggesting that a multimechanistic approach targeting both humoral and T cell-mediated immunities may be a key strategy to provide protective immunity and prevent new *S. aureus* infections [5]. In line with this, we identified that animals vaccinated with these *S. aureus* proteins associated with a GM-CSF DNA vaccine trigger type 3 cell-mediated immunity, reinforcing their potential use to prevent new IMIs by this pathogen.

It is well-known that T cells make an essential contribution toward *S. aureus* control [36,37,38]. In fact, T cell deficiencies increase the susceptibility to *S. aureus* infections [6,37]. Surprisingly, in developing an effective vaccine for bovine *S. aureus* mastitis, T cell responses have not received much attention, with unconventional T cells being especially neglected. In this setting, γδ T cells represent a largely ignored target for vaccine design. However, they comprise up to 60% of circulating lymphocytes in young cattle and up to 30% of blood mononuclear cells in adult animals [39]. The critical role of this T cell population in cattle could be envisaged, seeing that WC1^+^ γδ T cells are the first to arrive at the sites of mycobacterial purified protein derivate (PPD) injection and increase in frequency in blood following mycobacterial vaccination [39,40]. Moreover, γδ T cells present cytotoxic activity in cows, and a decrease of this cell population is associated with periods of increased susceptibility to IMIs [41].

Therefore, there is an increasing interest in studying the T cell response to *S. aureus* infections. Different subpopulations of T cells are likely to contribute to anti-staphylococcal immune defense. Among them, recent evidence has highlighted the importance of type 3 immunity for protective immunity against infections by *S. aureus* [10,11,12,13] and emphasizes their importance in developing new vaccines [10,12,42]. Type 3 immunity can be characterized by cells that encode IL-17A, IL-17F, IL-22 genes, and the Rorγt and Rorα transcription factors and their gene targets. The cells responsible for type 3 immunity are diverse, including γδ T cells, T CD4^+^ helper (Th17), T CD8^+^ (Tc17), and innate lymphoid cells 3 [7].

Among T cells producing IL-17, the nontraditional γδ T cells have been identified as a potent source of innate IL-17 and implicated in host protection in murine models of *S. aureus* infection. These findings reveal that γδ T cells are an important source of IL-17 in adaptive immunity and indicate that targeting the induction of nontraditional lymphocytes as specific γδ T cell subsets that secrete IL-17 represents a potentially important and novel target for the rational design of future vaccines against *S. aureus* [6,13]. The γδ T cells are known to substantially contribute to protective immunity against *S. aureus* infections involving innate and adaptive immune responses, as they act as the first line of defense and control the innate response by neutrophil-mediated regulation [41,43]. It has also been proposed that γδ T-lymphocytes play a crucial role in antibacterial immunity and may provide a unique barrier for mucosal microenvironments against bacterial pathogens [44]. Furthermore, it was recently shown that an increase in the subpopulation of γδ T cells (i.e., TCRVγ4^+^ cells) is essential for protection against subsequent infections by *S. aureus* [13,45,46]. Therefore, this T γδ lymphocyte subpopulation represents an important participant in the protective vaccine against *S. aureus* infections [12,13,45,47]. In fact, a central finding of the present study is the higher percentage of IL-17A^+^ TCRVγ4^+^ cells upon stimulation with *S. aureus*, which strongly indicates a pivotal protective role of our vaccination approach against *S. aureus* challenge infections. Further studies in vaccinated animals under intramammary challenge with *S. aureus* are certainly needed to confirm this assumption.

Interleukin-17 is associated with neutrophil mobilization, immune response modulation, antigen-specific inflammation [13,48], and mediating the communication between the immune system and mammary epithelial cells [49]. For example, epithelial cells may express receptors for IL-17 and IL-22. The exposure to IL-17 leads to the release of chemokines and granulocyte-macrophage colony-stimulating factor (GM-CSF), leading to neutrophil recruitment and activation, leading to the death of *S. aureus* [10,42,50]. The GM-CSF, in turn, activates the antigen-presenting cells, such as macrophages and dendritic cells, increasing major histocompatibility complex (MHC) expression, increasing its antigen-presenting capacity, and amplifying the primary antibody response. GM-CSF also stimulates the maturation of dendritic cells and their consequent migration towards regional lymph nodes. GM-CSF may therefore induce differentiation, proliferation, and activation of several cell types, facilitating the development of both humoral and cellular immunity [51]. We may thus conclude that GM-CSF is an important mediator of the immune response in the mammary glands of dairy cows [15,49,52,53], corroborating the importance of our findings and the use of GM-CSF DNA vaccination to improve vaccine efficacy against *S. aureus*.

The importance of γδ T cells is because they are the primary source of interleukin-17 (IL-17), while other non-γδ T cells only provide a small contribution for the production of IL-17 during *S. aureus* infections [13,29,45,46,54,55]. Our outcomes follow the same direction with γδ T cells, including their subpopulation TCRVγ4^+^ cells, having a critical role in IL-17A production, while αβ T cells only marginally produce IL-17A in the regional draining lymph nodes. Besides γδ T cells playing a pivotal contribution for IL-17A production by splenocytes with or without *S. aureus* stimulus in cell culture, Tc17 also contributes to type 3 cell-mediated immunity. Furthermore, γδ T cells, particularly TCRVγ4^+^ cells, robustly support IFN-γ production in the investigated draining lymph nodes. In contrast, both γδ T cells and Tc1 (T CD8^+^) cells markedly contributed to the IFN-γ when recall immune response was evaluated. The production of IFN-γ by γδ T cells, including TCRVγ4^+^ cells, has been previously demonstrated [47,56,57]. Overall, these findings support the substantial and broad role of this nonconventional T cell population in immunity against *S. aureus*.

Furthermore, we observed that the vaccination with the three recombinant proteins associated with the GM-CSF DNA vaccine downregulates T_H_2 immunity, observing decreased IL-4. Of note, a type 2 immune environment can promote *S. aureus* colonization and superinfection [29]. We have also found lower levels of IL-2 in the supernatant of splenocyte cell culture in animals that received the combination of *S. aureus* proteins together with the GM-CSF. Activation of T cells with IL-2 induces TEM cells, while T cells stimulated in vitro with IL-15 acquire a TCM phenotype [58]. However, as IL-15 was not determined, the latter remains speculative but somehow corroborates with the formation of the TCM phenotype found here. However, although the mechanism is still under debate, it should be noted that immediately after pathogen clearance the antigen-specific memory T-cell pool mainly comprises of TEM cells, which gradually convert to a TCM cell phenotype over time [59]. With this perspective, TCM cells are regarded as a renewable source of the T effector cells responsible for protection from acute infections and primed for a rapid effect response [60].

## 5. Conclusions

In the present study, immunization with the three recombinant proteins associated with GM-CSF DNA improved T cell memory response, especially γδ TCR^+^ and its subpopulation TCRVγ4^+^, which are majorly responsible for the production of IL-17A. Thus, as our vaccination approach fosters type 3 immunity, a protective immune response against *S. aureus* was expected. Therefore, although our outcomes are promising, further studies in *S. aureus* challenged the mice model and ruminants will be necessary to validate our findings regarding the protective immunity against *S. aureus*.

## 6. Patents

The authors have a patent titled “Vaccine composition, kit for diagnosis of *Staphylococcus aureus* infections in ruminants, method and uses”.

## Figures and Tables

**Figure 1 vaccines-09-00899-f001:**
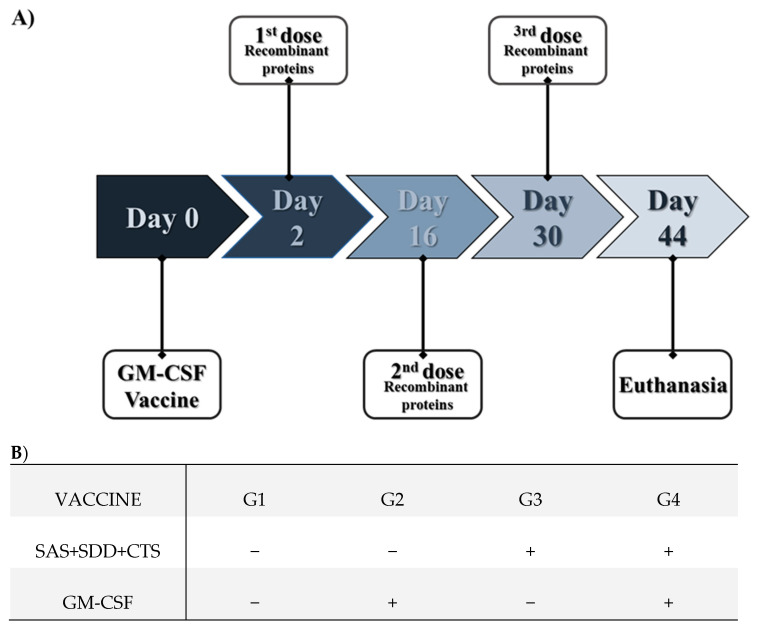
Scheme of the timeline of vaccination (**A**) and experimental groups (**B**). SAS: F0F1 ATP synthase subunit alpha recombinant *S. aureus* protein; SDD: succinyl-diaminopimelate desuccinylase recombinant *S. aureus* protein; CTS: cysteinyl-tRNA synthetase recombinant *S. aureus* protein; GM-SCF: pCI-granulocyte and macrophage colony-stimulating factor plasmid DNA (GM-CSF DNA vaccine). The unvaccinated group just received the liposome or saponin adjuvant.

**Figure 2 vaccines-09-00899-f002:**
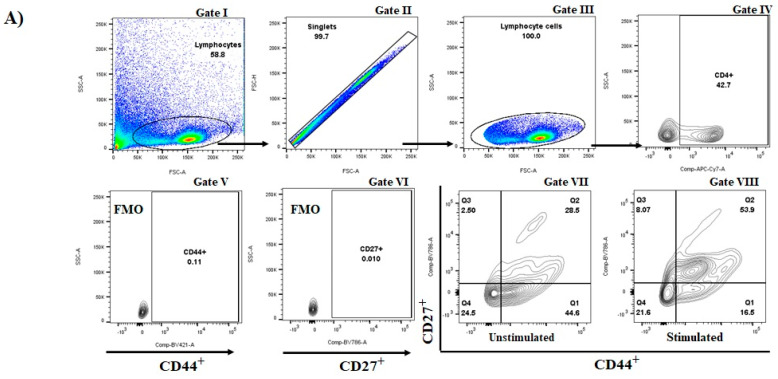
The antidromic trend of the percentage of T CD4^+^ CD44^+^ CD27^+^ cells and T CD8^+^ CD44^+^ CD27^−^ cells in draining lymph nodes from mice vaccinated with three-recombinant *S. aureus* proteins. (**A**) A representative gating hierarchy (successive gating scheme as depicted by arrows): lymphocytes (gate I), single cells are identified by plotting forward scatter area against forward scatter-height (gate II), single lymphocyte cells (gate III), T CD4^+^ cells (gate IV), FMO controls for CD44 (gate V) and CD27 (gate VI); and representative plots used to discriminate and identify two distinct T CD4^+^ memory cell populations: a CD44^high^ CD27^low^ population, and a CD44^high^ CD27^high^ population unstimulated (gate VII) and stimulated with *S. aureus* (gate VIII). (**B**) Percentage of T CD4^+^ CD44^+^ CD27^+^ cells and T CD8^+^ CD44^+^ CD27^−^ cells in draining lymph nodes among groups. FMO: fluorescence minus one; F0F1 ATP synthase subunit α (SAS), succinyl-diaminopimelate (SDD), and cysteinyl-tRNA synthetase (CTS). (G1, *n* = 6; G2, *n* = 6; G3, *n* = 6; G4, *n* = 6). G1: control; G2: Granulocyte Macrophage Colony-Stimulating Factor (GM-CSF) DNA vaccine; G3: F0F1 ATP synthase subunit α (SAS), succinyl-diaminopimelate (SDD), and cysteinyl-tRNA syn-thetase (CTS) recombinant proteins; and G4: SAS+SDD+CTS plus GM-CSF DNA vaccine. * *p* < 0.05; ** *p* < 0.01 (One-way ANOVA followed by Tukey test).

**Figure 3 vaccines-09-00899-f003:**
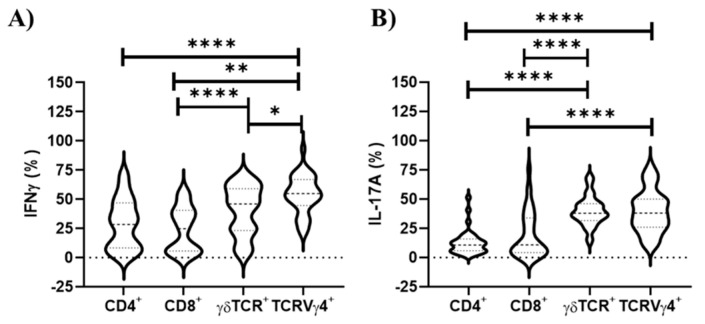
Cellular sources of IFN-γ and IL-17A in draining lymphocyte populations upon in vitro *S. aureus* stimulation. The γδ TCR cells and their subpopulation, TCRVγ4^+^ cells, are the primary sources of both IL-17A and IFN-γ in draining lymph nodes, rather than αβ T cells. (G1, G2, G3 and G4 combined; *n* = 24). G1: control; G2: Granulocyte Macrophage Colony-Stimulating Factor (GM-CSF) DNA vaccine; G3: F0F1 ATP synthase subunit α (SAS), succinyl-diaminopimelate (SDD), and cysteinyl-tRNA synthetase (CTS) recombinant proteins; and G4: SAS+SDD+CTS plus GM-CSF DNA vaccine. * *p* < 0.05; ** *p* < 0.01; **** *p* < 0.0001 (Kruskal–Wallis test followed by Dunn’s test).

**Figure 4 vaccines-09-00899-f004:**
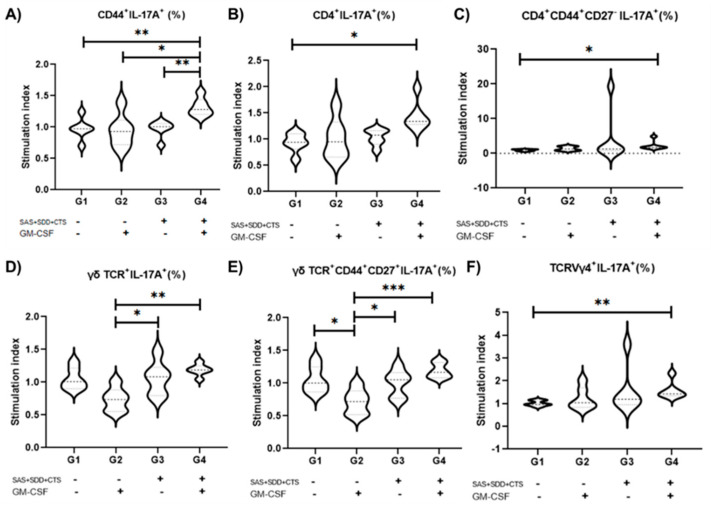
IL-17A^+^ T cells in mice immunized with three *S. aureus* recombinant proteins associated with the GM-CSF DNA vaccine. Data presented as stimulation index, normalizing T cell responses, dividing the percentage of IL-17A^+^ cells upon *S. aureus* stimulation by the percentage of IL-17A^+^ cells under unstimulated control condition. (**A**) CD44^+^ (memory) cells, (**B**) T CD4^+^ cells, (**C**) T CD4^+^ TEM (CD44^+^ CD27^−^) cells, (**D**) γδ TCR cells, (**E**) γδ TCR TCM, (**F**) TCRVγ4^+^ cells. (G1, *n* = 6; G2, *n* = 6; G3, *n* = 6; G4, *n* = 6). G1: control; G2: Granulocyte Macrophage Colony-Stimulating Factor (GM-CSF) DNA vaccine; G3: F0F1 ATP synthase subunit α (SAS), succinyl-diaminopimelate (SDD), and cysteinyl-tRNA syn-thetase (CTS) recombinant proteins; and G4: SAS+SDD+CTS plus GM-CSF DNA vaccine.* *p* < 0.05; ** *p* < 0.01; *** *p* < 0.001 [One-way ANOVA followed by Tukey test (**A**,**D**,**F**) and Kruskal–Wallis test followed by Dunn’s test (**B**,**C**,**E**)].

**Figure 5 vaccines-09-00899-f005:**
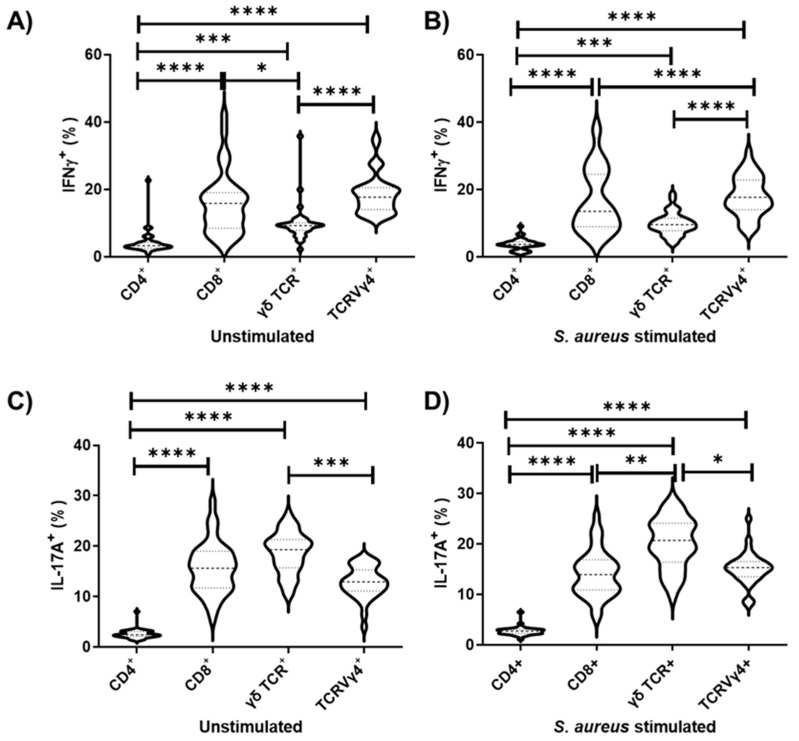
Cellular sources of IFN-γ (**A**,**B**) and IL-17A (**C**,**D**) in splenocyte populations upon in vitro *S. aureus* stimulation. The TCRVγ4^+^ cells and T CD8^+^ cells are the primary sources of IFN-γ, while γδ TCR is the major source of IL-17A, under splenocyte cell culture in unstimulated and *S. aureus* stimulated conditions. (G1, G2, G3 and G4 combined; *n* = 24). G1: control; G2: Granulocyte Macrophage Colony-Stimulating Factor (GM-CSF) DNA vaccine; G3: F0F1 ATP synthase subunit α (SAS), succinyl-diaminopimelate (SDD), and cysteinyl-tRNA syn-thetase (CTS) recombinant proteins; and G4: SAS+SDD+CTS plus GM-CSF DNA vaccine.* *p* < 0.05; ** *p* < 0.01; *** *p* < 0.001; **** *p* < 0.0001 [Kruskal–Wallis test followed by the Dunn’s test (**A**,**C**,**D**) and One-way ANOVA followed by Tukey test (**B**)].

**Table 1 vaccines-09-00899-t001:** Monoclonal antibodies (mAbs) used for immunophenotyping of splenocytes and lymph nodes lymphocytes by flow cytometry.

mAbs	Fluorescent Probes	Target	Clone	Host	Concentration (mg mL^−1^)	Cat. n.
Anti-CD4 ^1^	APC-Cy7	Mouse	GK1.5	Rat	0.2	552051
Anti-CD8 ^1^	BV510	Mouse	53-6.7	Rat	0.2	563068
Anti-CD44 ^1^	BV421	Mouse	IM7	Rat	0.2	563970
Anti-CD27 ^1^	BV750	Mouse	LG.3A10	Hamster	0.2	747399
Anti-CD19 ^1^	PE-Cy7	Mouse	1D3	Rat	0.2	552854
Anti-γδ TCR ^1^	BV650	Mouse	GL3	Hamster	0.2	563993
Anti-TCRVγ4 ^1^	FITC	Mouse	GL2	Hamster	0.2	552143
Anti-IL-17° ^1^	Alexa 700	Mouse	TC11-18H10	Rat	0.2	560820
Anti-IFN-γ ^1^	PE	Mouse	XMG1.2	Rat	0.2	554412

^1^ BD Pharmingen™ (San Diego, EUA); IL-17A: interleukin-17A; IFN-γ: interferon-γ; FITC: Fluorescein isothiocyanate; APC-Cy7: Allophycocyanin-cyanine 7; PE: R-phycoerythrin; PE-Cy7: R-Phycoerythrin-cyanine 7; BV510: Brilliant Violet 510; BV421: Brilliant Violet 421; BV650: Brilliant Violet 650; BV750: Brilliant Violet 750.

## Data Availability

The data presented in this study are available on reasonable request from the corresponding author.

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
