# Peer review of "Staphylococcus aureus Protection-Related Type 3 Cell-Mediated Immune Response Elicited by Recombinant Proteins and GM-CSF DNA Vaccine"

_vaccines, 2021, doi:10.3390/vaccines9080899_

Round 1

Reviewer 1 Report

In this manuscript, the authors described an immunization strategy of GM-CSF DNA vaccine followed by recombinant proteins to improve Staphylococcus Aureus protection-related type 3 cell-mediated immune response. However, more details need to be provided and some figures and descriptions should be modified.

  1. The immunogenicity of the strategies in naive mice is evaluated in this study, but no direct evidence to show the improvement of protective effect of combination strategy in an animal model (e.g., in an animal model suffered from Staphylococcus Aureus). The authors should provide more evidences to show the superiority of combination strategy (proteins plus DNA), compared with other groups.
  2. Provide some data to show sizes and purity of recombinant proteins, for example SDS-PAGE , WB or HPLC results.
  3. Show the results to demonstrate the GM-CSF expression mediated by the DNA vaccine in vivo or in vitro.
  4. Modify Figure 2A with images of high-resolution to make sure the percentage values can be seen clearly. Why the authors didn’t show the percentages of CD4+ CD44+ CD27- and CD8+ CD44+ CD27+ T cells in figure 2A? Did the cells in figure 2B were stimulated with S. aureus?
  5. Please clarify which group(s) was analyzed in Figure 3 and Figure 5.
  1. Line 321, “No effect of immunization on the induction of B cell response was revealed”. Please show some data as evidences.
  2. Please supply some data in the result 3.6.
  3. Please clarify the numbers of mice/samples involved for analysis in all related figure legend.

Author Response

Response to Reviewer 1:

First of all, thank you very much for kindly reviewing the manuscript and for the positive contribution to our manuscript.

Comments to the Author:

“The immunogenicity of the strategies in naive mice is evaluated in this study, but no direct evidence to show the improvement of protective effect of combination strategy in an animal model (e.g., in an animal model suffered from Staphylococcus aureus). The authors should provide more evidences to show the superiority of combination strategy (proteins plus DNA), compared with other groups.”

A.U.: The ultimate goal of this research line is a vaccine able to protect from S. aureus IMI as appointed by the reviewer. However, an initial study looking for promising immunogens and immunization schedules was needed, what we meant in the present study. Using an innovative approach through immunoproteomic using sera from supposedly immune cows, we identified S. aureus proteins that have not been studied before. Further, these proteins in combination with GM-CSF DNA vaccine induced a type 3 immune response suggested as effective in the protective immune response to IMI. Further study is planned to use these antigens with GM-CSF vaccine to evaluate the protection in mice that will receive in vivo mammary gland challenge with S. aureus. Then we expect to show the superiority of our approach compared with other known vaccine or vaccine candidates.

Comments to the Author:

“Provide some data to show sizes and purity of recombinant proteins, for example SDS-PAGE, WB or HPLC results.”

A.U.: Thank you for this valuable comment. The production of the recombinant proteins was performed, and their sizes and purity were assessed by SDS-PAGE (Figure 1; please, see the attached file). This information was included in the revised manuscript (Lines 114 and 115).

Figure 1. SDS-PAGE analysis of Staphylococcus aureus recombinant proteins. Coomassie blue-stained 12% SDS-PAGE analysis of cell lysate insoluble fraction and purified fractions resulting from IPTG-induced bacterial cultures showing bands in the expected sizes for S. aureus recombinant proteins. MM: molecular mass in kDa.  Bands of approximately 52.5 kDa, 56.2 kDa and 52.7 kDa referring to succinyl-diaminopimelate (SDD), F0F1 ATP synthase subunit α (SAS) and cysteinyl-tRNA synthetase (CTS) proteins were observed, respectively. SDS-PAGE: Polyacrylamide Gel Electrophoresis; IPTG: Isopropyl β-d-1-thiogalactopyranoside. 

Comments to the Author:

“Show the results to demonstrate the GM-CSF expression mediated by the DNA vaccine in vivo or in vitro.”

A.U.: Thank you for this comment. We set up an experiment to demonstrate the in vitro GM-CSF expression induced by the GM-CSF DNA vaccine transfecting RAW 264.7 cells (a murine macrophage cell line) with this DNA vaccine using liposome. We followed the previously established protocol by our group in HEK 293T/17 human cells (Fotoran et al., Mol. Ther. - Methods Clin. Dev. 2017, 7, 1–10, doi: 10.1016/j.omtm.2017.08.004.) with some modifications.

Briefly, the RAW 264.7 cells were transfected with GM-CSF DNA vaccine using the liposome in triplicate. The total RNA was extracted from 2 x 106 transfected cells mL-1, and 1 μg of the total RNA was reverse transcribed using High Capacity cDNA Reverse Transcription Kit reagent (Applied Biosystems, Foster City, CA, USA) to obtain the cDNA, as recommended by the manufacturers. Further, the expression of the GM-CSF mRNA in RAW 264.7 cells was determined by the qPCR technique. For this assay, specific primers for the optimized DNA vaccine sequence (Table 1) used in our study were designed. These primers are specific to our GM-CSF DNA vaccine and have no similarity with mouse GMCSF mRNA and other mouse genes. We would emphasize that our DNA vaccine used an optimized sequence that differs from mice GM-CSF mRNA (with different nucleotide sequence, but with no difference in the amino acids sequence). The nucleotide sequences of the mouse GM-CSF vaccine primers used were: forward TGCTGTTTCTGGGCATTGTCGTGT, and reward: ATCGCTTCCACATGTTTCCACGGA).  Subsequently, 1 ug of cDNA was mixed HOT FIREPol EvaGreen qPCR mix Plus (cat. n. 08-24-00001, Solys BioDyne, Tartu, Estonia), and the qPCR reaction was performed according to the manufacturer's instructions, in StepOne PCR system (Applied Biosystems, Foster City, CA, USA) using the following conditions: 95°C for 12 min, 40 cycles of 92 °C for 15 sec, 61 °C for 30 sec, and 70 °C for 30 sec. After amplification by qPCR, a melting curve was generated to confirm the specificity of the product.

Here, we show in Figure 2 (please, see the attached file) a single peak in the melting curve (80 °C) observed in RAW 264.7 cells transfected with the GM-CSF DNA vaccine.

Table 1. Nucleotide sequence of the GM-CSF DNA vaccine

Optimized GM-CSF vaccine sequence

GGATCCC CCGGG ATG TGG CTG CAG AAT CTG CTG TTT CTG GGC ATT GTC GTG TAT TCC CTT TCA GCA CCT ACT CGC AGT CCG ATT ACG GTA ACT CGT CCG TGG AAA CAT GTG GAA GCG ATC AAA GAA GCG TTG AAC CTG TTA GAC GAT ATG CCG GTT ACC CTC AAT GAG GAA GTC GAA GTA GTG TCT AAC GAG TTT AGC TTC AAG AAA CTG ACG TGT GTG CAA ACC CGC TTG AAG ATC TTC GAA CAA GGG TTA CGT GGC AAC TTC ACC AAA CTG AAA GGT GCC CTT AAT ATG ACC GCT TCG TAC TAC CAG ACC TAT TGT CCG CCA ACT CCA GAA ACC GAT TGC GAA ACA CAG GTT ACC ACG TAT GCC GAC TTT ATC GAT AGC CTC AAA ACG TTT CTG ACA GAT ATT CCG TTT GAG TGC AAG AAA CCC GGT CAG AAA GTCGACGAATTC

GM-CSF: Granulocyte Macrophage Colony-Stimulating Factor

Comments to the Author:

“Modify Figure 2A with images of high-resolution to make sure the percentage values can be seen clearly. Why the authors didn’t show the percentages of CD4+ CD44+ CD27- and CD8+ CD44+ CD27+ T cells in figure 2A? Did the cells in figure 2B were stimulated with S. aureus?”

A.U.: Thank you for this comment. We modified Figure 2A as suggested. We believe that Figure 2B referred by the reviewer is that on the CD4+ CD44+ CD27- and CD8+ CD44+ CD27+ data. As no statistical difference was found in the stimulation index of the percentage of CD4+ CD44+ CD27- (P = 0.27) and CD8+ CD44+ CD27+ (P = 0.75), they were not included in the Figure 2A. Nonetheless, the information regarding the non-significant difference of these populations was now included in the revised manuscript (Lines 288-289). We included Figure 3 revision note; please, see the attached file) with our data regarding these lymphocytes populations for reviewer information. Furthermore, we must highlight that Figure 2 is related to non-stimulated cells (recall immunity), as it refers to cells of the draining lymph nodes (unfortunately, we do not have enough cells to perform cell culture of the draining lymph nodes under S. aureus challenge conditions).

Comments to the Author:

Please clarify which group(s) was analyzed in Figure 3 and Figure 5.

A.U.: In both figures (Figures 3 and 5), we analyzed all animals (animals from G1, G2, G3, and G4 combined) to assess the impact of distinct lymphocyte populations on the intracellular production of interferon-γ and interleukin-17A (both the percentage and the intensity of cytokines production). Therefore we analyzed the capacity of each T lymphocyte population to produce these cytokines under stimulation of S. aureus originated from a persistent bovine intramammary infection, as shown by Murphy et al. (Immunol. 2014, 192, 3697–3708, doi:10.4049/jimmunol.1303420) using a well-characterized PS80 S. aureus penicillinase-producing strain (Asheshov, Journal of General Microbiology 1969, 59, 289–301, doi:10.1099/00221287-59-3-289). This information is now clearly included in the revised manuscript (Lines 321 and 378). G1: control; G2: Granulocyte-Macrophage Colony-Stimulating Factor (GM-CSF) DNA vaccine; G3: F0F1 ATP synthase subunit α (SAS), succinyl-diaminopimelate (SDD), and cysteinyl-tRNA synthetase (CTS) recombinant proteins; and G4: SAS+SDD+CTS plus GM-CSF DNA vaccine.

Comments to the Author:

“Line 321, “No effect of immunization on the induction of B cell response was revealed”. Please show some data as evidences”.

A.U.: Thank you for this consideration. In our study, the B lymphocyte population showed no statistically significant difference, and this information is now included in the revised manuscript (Lines 334-336). For reviewer information, the stimulation index of B lymphocytes (CD19+) population (P = 0.21) (Figure 4; please, see the attached file).

Comments to the Author:

“Please supply some data in the result 3.6.”

A.U.: Thank you for the consideration. As mentioned in the manuscript, we observed that animals that received the S. aureus recombinant proteins plus GM-CSF DNA vaccine led to an increase in the percentage of proliferative (ki67+) overall memory T cells (CD44+; P = 0,02), CD4+ cells (P = 0.02) and CD4+ T effector memory cells (CD44+ CD27-; P = 0.01), as demonstrated  (Supplementary Figure 3; please, see the attached file). We also included this figure as a supplementary file (Supplementary Figure 3) in the revised manuscript. No significant difference when stimulated with S. aureus was found in the percentage of proliferative (ki67+) overall memory T cells (CD44+; G1: P = 0.62; G2: 0;19; and G3 = 0.20), CD4+ cells (G1: P = 0.87; G2: 0.27; and G3 = 0.28) and CD4+ TEM (G1: P = 0.60; G2: 0.33; and G3 = 0.16) in the other groups.

Comments to the Author:

Please clarify the numbers of mice/samples involved for analysis in all related figure legend.

A.U.: Thank you for this suggestion. The number of animals used was clearly added in the revised manuscript (Lines 302, 321, and 359).

Sincerely yours,

Kamila Reis Santos and co-authors

Veterinary Clinical Immunology Research Group

University of São Paulo, Brazil

Reviewer 2 Report

Dear Editors!

In the manuscript with the title "Staphylococcus Aureus Protection-Related Type 3 Cell-Medi-2 ated Immune Response Elicited by Recombinant Proteins and 3 GM-CSF DNA Vaccine" the authors investigated the immunity elicited by different vaccines containing either artificially expresses proteins of S. aureus and/or a DNA vaccine encoding GM-CSF. The experiment was well designed and performed in mice. The authors prepared the results section very well and understandable also for non-immunologists. The different vaccination schemes resulted in different outcomes, nonetheless the combination of the proteins and the DNA vaccine could elicite a type 3 immunity represented by IL-17A expression. The authors claim, that this type of immunity may be promising not only in mice, but also in dairy herds, which suffer from S. aureus mastitis.

Overall, this manuscript represents a well designed study and well presented study results. The discussion is fine and the conclusion section is sound.

I added my few comments and corrections directly in the pdf version of the manuscript!

I absolutely recommend to publish this quality manuscript in Vaccines!

Author Response

Response to Reviewer 2:

First of all, thank you very much for kindly reviewing the manuscript and for the positive comments on our paper.

Comments to the Author:

“In the manuscript with the title "Staphylococcus Aureus Protection-Related Type 3 Cell-Mediated Immune Response Elicited by Recombinant Proteins and 3 GM-CSF DNA Vaccine" the authors investigated the immunity elicited by different vaccines containing either artificially expresses proteins of S. aureus and/or a DNA vaccine encoding GM-CSF. The experiment was well designed and performed in mice. The authors prepared the results section very well and understandable also for non-immunologists. The different vaccination schemes resulted in different outcomes, nonetheless the combination of the proteins and the DNA vaccine could elicite a type 3 immunity represented by IL-17A expression. The authors claim, that this type of immunity may be promising not only in mice, but also in dairy herds, which suffer from S. aureus mastitis.

Overall, this manuscript represents a well designed study and well presented study results. The discussion is fine and the conclusion section is sound.

I added my few comments and corrections directly in the pdf version of the manuscript!

I absolutely recommend to publish this quality manuscript in Vaccines!

A.U.: Thank you for your encouraging comments. We appreciate all your comments about our work. All suggestions were accepted and modified in the text.

Sincerely yours,

Kamila Reis Santos and co-authors

Veterinary Clinical Immunology Research Group

University of São Paulo, Brazil

Round 2

Reviewer 1 Report

The response is  very detailed. 

Author Response

Thank you for kindly dealing with our manuscript and for the overall positive contribution.